# Mimicking the Intestinal Host–Pathogen Interactions in a 3D In Vitro Model: The Role of the Mucus Layer

**DOI:** 10.3390/pharmaceutics14081552

**Published:** 2022-07-26

**Authors:** María García-Díaz, Maria del Mar Cendra, Raquel Alonso-Roman, María Urdániz, Eduard Torrents, Elena Martínez

**Affiliations:** 1Institute for Bioengineering of Catalonia (IBEC), The Barcelona Institute of Science and Technology (BIST), 08028 Barcelona, Spain; mar.cendra@upc.edu (M.d.M.C.); raquel.roman@leibniz-hki.de (R.A.-R.); murdaniz@ibecbarcelona.eu (M.U.); etorrents@ibecbarcelona.eu (E.T.); 2Microbiology Section, Department of Genetics, Microbiology and Statistics, Biology Faculty, University of Barcelona, 08028 Barcelona, Spain; 3Centro de Investigación Biomédica en Red de Bioingeniería, Biomateriales y Nanomedicina (CIBER-BBN), 28029 Madrid, Spain; 4Department of Electronics and Biomedical Engineering, University of Barcelona, 08028 Barcelona, Spain

**Keywords:** 3D in vitro models, intestinal models, host–pathogen interaction, intestinal mucus, hydrogels

## Abstract

The intestinal mucus lines the luminal surface of the intestinal epithelium. This mucus is a dynamic semipermeable barrier and one of the first-line defense mechanisms against the outside environment, protecting the body against chemical, mechanical, or biological external insults. At the same time, the intestinal mucus accommodates the resident microbiota, providing nutrients and attachment sites, and therefore playing an essential role in the host–pathogen interactions and gut homeostasis. Underneath this mucus layer, the intestinal epithelium is organized into finger-like protrusions called villi and invaginations called crypts. This characteristic 3D architecture is known to influence the epithelial cell differentiation and function. However, when modelling in vitro the intestinal host–pathogen interactions, these two essential features, the intestinal mucus and the 3D topography are often not represented, thus limiting the relevance of the models. Here we present an in vitro model that mimics the small intestinal mucosa and its interactions with intestinal pathogens in a relevant manner, containing the secreted mucus layer and the epithelial barrier in a 3D villus-like hydrogel scaffold. This 3D architecture significantly enhanced the secretion of mucus. In infection with the pathogenic adherent invasive *E. coli* strain LF82, characteristic of Crohn’s disease, we observed that this secreted mucus promoted the adhesion of the pathogen and at the same time had a protective effect upon its invasion. This pathogenic strain was able to survive inside the epithelial cells and trigger an inflammatory response that was milder when a thick mucus layer was present. Thus, we demonstrated that our model faithfully mimics the key features of the intestinal mucosa necessary to study the interactions with intestinal pathogens.

## 1. Introduction

The human small intestine is a complex organ primarily responsible not only for nutrient absorption but also for acting as a barrier to the outside environment. The intestinal epithelium is organized in a three-dimensional (3D) complex topography formed by invaginations called crypts and finger-like protrusions called villi. This epithelial layer contains multiple differentiated intestinal epithelial cells including absorptive enterocytes and mucus-secreting goblet cells. The mucus layer covering the intestinal epithelium harbors a complex bacterial community, referred to as the intestinal microbiota, while providing the first line of defense against external insults and pathogens to preserve homeostasis [1,2]. This mucus layer is a viscoelastic hydrophilic gel mainly composed of water, mucins, and small quantities of lipids and proteins [3,4]. Mucins are highly glycosylated proteins that form an entangled network, acting as a protective and selective barrier preventing microorganisms from reaching the epithelial surface. At the same time, these glycans are used as ligands for bacterial adhesion and as a nutrient source, thus providing a niche for bacterial colonization [5,6]. Some pathogens have evolved specialized strategies to successfully penetrate and exploit the mucosal and cellular barriers to infection [7]. In addition, patients suffering from certain pathologies such as the inflammatory bowel diseases have a mucus layer with an altered glycosylation profile, making it thinner and more penetrable to bacteria [8]. Thus, the interactions between the intestinal bacteria with the mucosal barrier play a crucial role in the regulation of the intestinal homeostasis and in the development of some pathologies such as the mentioned intestinal bowel disease or colorectal cancer [9]. However, fundamental knowledge of the mechanisms for these complex interactions remains limited due to the lack of relevant models that contain the main key players involved.

The use of genetically engineered murine models have been instrumental in deciphering the role of mucins in protecting the intestinal epithelium and in the development and pathogenesis of intestinal inflammatory events [10,11]. However, in vivo studies are restricted to end-point measurements, and it is difficult to assess the dynamic interactions between pathogens and the mucosal barrier. On the other hand, the traditional in vitro models based on two-dimensional (2D) tissue cultures often underrepresent the complexity of the intestinal mucosal environment, limiting the relevance of the data obtained. To circumvent some of these drawbacks, a variety of advanced 3D in vitro models have been developed in recent years to better recapitulate the host–microbiome crosstalk in the human gastrointestinal tract [12,13]. The 3D topography has a tremendous effect on cell behavior of epithelial barriers [14]. Specifically, intestinal epithelial cells have been shown to respond to the villus-like topography promoting cell differentiation and mucin secretion [15,16].

On the other hand, bacterial colonization in the small intestine greatly depends on the 3D topography, the mucus layer, and the oxygen gradients generated along the crypt-villus axis [17]. In addition, the intestinal mucosa is exposed to the intestinal luminal fluid, a dynamic mixture of enzymes, lipids and bile salts that is essential for the digestion and absorption of nutrients. The effect of this intestinal fluid on the solubility and permeability of molecules is known, but it also plays a role in how microbial communities interact [18].

In this work, we used a 3D engineered intestinal model to study the role of the intestinal mucus on the interactions with intestinal microbiota, in a simple but yet physiologically relevant manner. This engineered model has some of the key elements for the study of these host–microorganism interactions: the 3D topography mimicking the small intestine villus-like morphology, the intestinal epithelium representing the two most abundant cell types (enterocytes and mucus-secreting goblet cells); the secreted mucus layer; and a physiologically relevant intestinal fluid. The villus-like hydrogel scaffold was fabricated by a mold-less technique based on photolithography using poly (ethylene glycol) diacrylate (PEGDA) [19]. Cells grown in these 3D hydrogel scaffolds experience the physiological dimensions, mechanical properties, and curvature found in the human small intestine, and this influences the cell morphology and tissue barrier properties, with values closer to the in vivo situation than conventional monolayers [19,20]. In addition, these scaffolds are assembled to Transwell inserts, maintaining the compatibility with standard cell culture assays. We compared two different models: the monoculture of enterocyte-like Caco-2 cells and their co-culture with goblet-like HT29-MTX cells grown on top of the 3D villus hydrogels. The Caco-2 is the most used cell line for modeling the small intestine and the gold standard model for permeability drug screening; whereas its co-culture with HT29-MTX cells allows for the secretion of mucins predominantly expressed in the intestinal tract. As controls, we have their 2D counterpart models, grown on flat porous membranes. We infected these models with two strains of *E. coli*, the pathogenic adherent invasive LF82 (AIEC) and the commensal K-12 sub-strain MG1655.

The LF82 is one of the reference strains identified to have an important prevalence in patients with Crohn’s disease (CD) [21]. These strains have the ability to adhere to and invade intestinal epithelial cells, and gave rise to a new specific pathogenic group of *E. coli* called adherent-invasive *E. coli* (AIEC) [22,23,24]. The AIEC LF82 penetrates the mucus barrier by promoting mucin degradation with proteases [25]. After crossing the intestinal mucus layer, AIEC strains adhere and invade the epithelial cells via interaction with the cell adhesion molecule 6 (CEACAM6) receptors of the enterocytes [26]. For the non-pathogenic control, we used the *E. coli* MG1655, a standard laboratory strain that is not able to invade epithelial cells [27].

In this work, we demonstrate that 3D topography has a significant effect on the secretion of the intestinal mucus and that this mucus modulates the interactions between the intestinal pathogens and the epithelium. When the mucus is present, pathogens are able to invade less, and therefore the cellular response that is triggered is milder. Additionally, these host–pathogen interactions can be influenced by the luminal intestinal fluid. Thus, our work emphasizes the importance of including all the key players involved in the intestinal mucosa microenvironment to provide a realistic insight into the cells’ and microorganisms’ behavior.

## 2. Materials and Methods

### 2.1. Materials

Poly (ethylene glycol) diacrylate (PEGDA) (MW 6000 g/mol), 2-hydroxy-4′-(2-hydroxyethoxy)-2-methylpropiophenone (Irgacure D-2959), acrylic acid (AA), 1-ethyl-3-(3-dimethylaminopropyl) carbodiimide hydrochloride (EDC), N-hydroxysuccinimide (NHS), collagen type I, 4 kDa fluorescein-isothiocyanate dextran (FD4) and sodium taurocholate hydrate were purchased from Sigma-Aldrich (St. Louis, MO, USA). Phosphate buffered saline (PBS) was from ThermoFisher (Waltham, MA, USA). Polydimethylsiloxane (PDMS) Sylgard 184 was purchased from Dow Corning (Midland, MI, USA). Soybean L-α-phosphatidylcholine (95%) was purchased from Avanti Polar Lipids (Alabaster, AL, USA).

### 2.2. Fabrication of Villus-like Hydrogel Scaffolds

The microstructured scaffolds were fabricated using a single-step and mold-less technique based on dynamic photopolymerization as shown in Figure 1A and previously described in [19]. Briefly, the prepolymer solution of 6.5% *w*/*v* PEGDA, 0.3% *w*/*v* AA and 1% *w*/*v* Irgacure D-2959 was prepared in PBS and flown into a PDMS chip with an array of pools of 6.5 mm diameter. Pools were then covered by porous membranes (Tracketch^®^ polyethylene terephthalate (PET), 5 μm pore size), which were the substrate holders for the microstructured hydrogel scaffolds. Subsequently, the chip was then exposed to UV using patterned photomasks. After light exposure, the villus-like hydrogel scaffolds were stored in PBS for minimum 3 days to reach equilibrium swelling. Then, samples were mounted on standard 24-well Transwell^®^ inserts using double-sided pressure-sensitive adhesive (PSA) rings [19] under sterile conditions (Figure 1B). Prior to cell seeding, villus-like scaffolds were functionalized with collagen type I using EDC/NHS coupling chemistry, providing the PEGDA-AA scaffolds with cell-adhesion motifs.

### 2.3. Cell Culture

Caco-2 cells (passage 74–80) and HT29-MTX cells (passage 35–42) were expanded separately and maintained in 75 cm^2^ flasks in high glucose DMEM (Gibco, Thermofisher, Waltham, MA, USA), supplemented with 10% *v*/*v* fetal bovine serum (FBS) (Gibco, Thermofisher), 1% *v*/*v* penicillin/streptomycin (Sigma-Aldrich), and 1% *v*/*v* non-essential amino acids (Gibco, Thermofisher). Cell cultures were maintained in an incubator at 37 °C and 5% CO_2_, changing the medium every three days, and passaged weekly. For the experiments, Caco-2 cells or the co-culture of Caco-2 and HT29-MTX cells were mixed at 90:10 ratio and seeded on the 3D hydrogels mounted on the modified inserts at a final density of 2.5 × 10^5^ cells/cm^2^. Two-dimensional control experiments were performed on cells cultured in standard 24-well polycarbonate Transwell^®^ filter inserts (0.33 cm^2^ growth area, 0.4 μm pore size) at a final density of 1.5 × 10^5^ cells/cm^2^. Cells were maintained for 21 days as the standard for Caco-2 cell polarization, exchanging media every other day. In the case of Caco-2/HT29-MTX co-culture, cells were maintained under submerged conditions during the first week after seeding, and then changed to an air–liquid interface culture for the following two weeks, exchanging the media at the lower compartment every other day.

### 2.4. Intestinal Epithelial Barrier Characterization

Transepithelial electrical resistance (TEER) was monitored every other day with an EVOM2 Epithelial voltohmmeter with an EndOhm-6 chamber (World Precision Instruments). TEER values were normalized by the total surface area of the cell monolayer. After 21 days, drug permeability studies were performed using FITC-dextran of 4.4 kDa (FD4) as paracellular marker. Briefly, cells were washed with PBS before adding 200 μL of 0.5 mg/mL FD4 to the apical compartment and 600 μL DMEM without phenol red to the basolateral compartment. Throughout the experiment, cells were incubated at 37 °C. At specific time points, samples were withdrawn from the basolateral compartment followed by media replacement. Retrieved samples were then transferred to a 96-well black plate and their fluorescence was read on a microplate reader (Infinite M200 PRO Multimode, Tecan, Männedorf, Switzerland) at 495/520 nm excitation/emission wavelengths. The apparent permeability coefficient (*P_app_*) was calculated by the following equation,
$$P_{app}=\frac{dQ}{dt}\cdot \;\frac{1}{A\cdot C_{0}}$$
where *dQ*/*dt* is the flux, *A* the surface area of the cell culture, and *C*_0_ the initial donor concentration of FD4. The steady state flux was used for the calculation of the *P_app_* value. All experiments were done in at least three independent biological replicates.

### 2.5. Cell Morphological Evaluation

After 14 or 21 days of culture, cells grown on the PEGDA-AA scaffolds were fixed with 10% neutral buffered formalin solution (Sigma-Aldrich) for 30 min at room temperature or with Carnoy’s solution for mucus fixation. In order to obtain high-resolution images along with the villus-like structures, histological cross-sections were obtained following a novel embedding method developed in our lab [20]. Briefly, the 3D villus-like microstructures were first embedded into a hydrogel block using a solution of 10% *w*/*v* 575 Da PEGDA (P575) and 1% *w*/*v* Irgacure D-2959 photoinitiator in PBS. To that end, the microstructured sample was placed in a PDMS pool of 10 mm in diameter and 2 mm in height and filled with the P575 embedding media. After UV exposure (100 s each side), a hydrogel block was obtained, containing the microstructured scaffold. This block was then re-embedded into OCT and cryotome sectioning was performed to obtain thin cross-sections with fully preserved villus-like microstructures. The cryosections were subjected to routine Hematoxylin–Eosin (HE) staining and visualized using bright field microscopy (Eclipse Ts2, Nikon, Melville, NY, USA). Cell polarization was evaluated by immunostaining of epithelial cell markers. Antigen retrieval was performed by boiling the cryosections in 10 mM citrate buffer and 0.05% Tween 20 at pH 6.0 for 10 min in a microwave oven. Then, samples were permeabilized with 0.5% Triton X-100 for 30 min and blocked with 1% bovine serum albumin (BSA) and 3% donkey serum. Primary antibodies against villin (5 μg/mL, Abcam ab201989, Cambridge, UK), ZO-1 (2 μg/mL, Abcam ab190085), and β-catenin (5 μg/mL, Abcam ab2365) were incubated for 72 h at 4 °C to allow better penetration through the hydrogel embedding block. Then, samples were incubated with the secondary antibodies Alexa Fluor 488 donkey anti-mouse (Invitrogen A-21202, Waltham, MA, USA), Alexa Fluor 647 donkey anti-rabbit (Jackson ImmunoResearch 111-607-003, West Grove, PA, USA), and Alexa Fluor 568 donkey anti-goat (Invitrogen A-11057) diluted at 4 μg/mL for 24 h at 4 °C. Nuclei were stained with 4’, 6-diamidino-2-phenylindole (DAPI) (5 μg/mL, Invitrogen D1306) for 1 h. For the staining of filamentous actin (F-actin), no antigen retrieval was performed as it interfered with the staining. Then, after permeabilization and blocking, samples were incubated with Acti-stain 535 Phalloidin (100 nM, Tebu-bio) for 2 h and counterstained with DAPI for 1 h. Samples were then mounted using Fluoromount-G (SouthernBiotech, Birmingham, AB, USA) and fluorescence images were acquired using a confocal laser scanning microscope (LSM 800, Zeiss, Oberkochen, Germany). Images were processed by Image J software (US NIH, Bethesda, MD, USA, https://imagej.nih.gov/ij/, accessed on 23 June 2022).

### 2.6. PAS Staining

Mucus secretion was visualized using the Periodic Acid-Schiff’s (PAS) assay that stains mucin glycoproteins. At different time points of culture, samples were fixed in Carnoy’s solution. Samples were then sequentially incubated with periodic acid and Schiff’s reagent (Sigma-Aldrich) at room temperature, for 5 and 15 min, respectively. Samples were then imaged using a confocal laser scanning microscope in the brightfield mode (LSM 800, Zeiss), acquiring *z*-stacks at 5 μm intervals. Images were processed using the Extended Depth of Field plugin of Image J software.

### 2.7. Bacterial Strains and Growth Conditions

Herein, we used the adherent invasive *Escherichia coli* (AIEC) LF82 strain [28] and the commensal *Escherichia coli* K-12 MG1655 strain (laboratory collection). Overnight (O/N; ~16 h) cultures of *E. coli* were grown in Luria-Bertani medium (LB; Scharlab, Barcelona, Spain) at 37 °C and 200 rpm of shaking. LB–agar (Scharlab) was used to count the respective bacterial colony forming units (CFU) of the *E. coli* strains. Bacterial growth was measured as optical density at 550 nm (OD_550_).

### 2.8. Bacterial Adhesion and Invasion Assays

Bacterial adhesion and invasion experiments were performed on the 3D and 2D models at day 21 after cell seeding. Before bacterial infection, cells were cultured with fresh antibiotic-free medium supplemented with 10% FBS and 1% non-essential amino acids for 24 h. O/N cultures of *E. coli* were washed twice with PBS. Cells were then infected through the apical compartment at a multiplicity of infection (MOI) of 100 (approximately 100 bacteria per epithelial cell). After 3 h of incubation at 37 °C, cells were washed three times with warm sterile PBS, and lysed with PBS containing 0.1% *w*/*v* of saponin (saponin buffer) for 15 min. Serial dilutions were plated onto LB–agar plates to determine the number of CFU of the total number of cell-adhered bacteria, which correspond to both adherent and intracellular bacteria.

Bacterial invasion was quantified using the gentamicin protection assay as previously used [29]. Briefly, after 3 h of infection, cells were washed three times with warm sterile PBS, and incubated with fresh cell culture medium supplemented with 200 µg/mL of gentamicin (gentamicin solution) for 90 min to kill extracellular bacteria. Cells were then lysed with saponin buffer, and intracellular bacteria enumerated as described above for the adherent bacteria. To measure the bacterial intracellular survival and replication overtime, infected cells were incubated for 24 h with gentamicin solution, and intracellular CFU of *E. coli* were determined at the time-point.

### 2.9. Scanning Electron Microscopy

After bacterial infection, 3D samples were rinsed with PBS and fixed with 2.5% glutaraldehyde for 1 h. Then samples were washed thoroughly with distilled water to remove the salts and performed serial dehydration in ethanol. Samples were then dried using a critical point drier and imaged using a scanning electron microscope (NOVA NanoSEM 230, FEI Company, Hillsboro, OR, USA).

### 2.10. Effect of Bacterial Infection on the Intestinal Barrier

The barrier integrity of the cell monolayers was evaluated before, immediately after bacterial infection, and 24 h post-infection (gentamicin treatment). TEER was measured with an EVOM2 Epithelial voltohmmeter with STX3 electrodes (World Precision Instruments) after 15 min equilibration at room temperature. In order to measure the interleukin IL-8 secreted by the epithelial cells after infection, apical supernatants were harvested and stored at −20 °C until further use. Samples were analyzed by ELISA (Human IL-8 ELISA kit, Invitrogen) following the manufacturer’s protocol.

### 2.11. Simulated Intestinal Fluid (SIF)

SIF was prepared by dissolving 5 mM sodium taurocholate and 1.25 mM L-a-phosphatidylcholine in DMEM supplemented with 10% FBS. The effect of the SIF on cell viability was analyzed with the CellTiter 96 Aqueous One Solution Proliferation Assay (Promega, Madison, WI, USA). SIF was added to the apical compartment of the 3D co-culture model and incubated for 3 h. Cells were then washed with PBS and membranes were detached from the Transwell inserts and placed in a 24-well plate. Then, cells were incubated with the kit reagents for 1 h, following the manufacturer’s protocol. The supernatant was then transferred to a 96-well plate and absorbances was read at 490 nm with an Infinite M200 PRO Multimode plate reader (Tecan). In order to account for the subproducts resulting from bacterial metabolism, SIF was incubated in the presence of *E. coli* LF82 for 3 h, mimicking the conditions of the infection experiments. Then, the metabolized medium was centrifuged to remove the bacteria and the supernatant filtered through 0.22 μm pore size prior to the incubation with the 3D co-culture model. The bacterial growth kinetics in the presence of SIF were also tested. Each bacterial overnight with a 1:100 dilution was inoculated in SIF, LB Broth or DMEM without phenol red. Bacteria were incubated at 37 °C, shaking at 200 rpm. A sample was taken every 20 min and optical density (OD) was measured at 550 nm. Prior to the bacterial infection, LF82 was incubated with SIF overnight in order to promote the expression of virulence factors. Then, adhesion and invasion assays were performed as stated above.

### 2.12. Statistics

Each experiment was performed in at least three independent replicates, with at least two cell culture wells for each condition. Statistical analyses were performed with GraphPad Prism (GraphPad software, La Jolla, CA, USA) using the unpaired *t*-test. A *p* value < 0.01 was considered significant.

## 3. Results

### 3.1. Cells Grown on 3D Villus-like Scaffolds Mimic the Native Intestinal Mucosal Barrier

The microstructured hydrogel scaffolds mimicking the intestinal villi were fabricated by photolithography-based dynamic photopolymerization as described in [19] (Figure 1A,B). This single-step and mold-less procedure yields microstructured hydrogels of about 400 μm in height, with similar anatomical dimensions and morphology of the intestinal villi (Figure 1C). A monoculture of Caco-2 cells or a mixture of Caco-2/HT29-MTX cells (ratio 90:10) were seeded on top of these 3D hydrogel scaffolds and cultured for 21 days (Figure 1D). In both cases, cells formed an epithelial monolayer with effective barrier properties, as TEER increased with culture time (Figure 1E). However, these TEER values are significantly lower than those reported for the conventional monolayers grown on flat porous membranes (Appendix A). As observed in our previous works, the tight junctions are significantly affected by the 3D topography and curvature of the villus-like hydrogel [19,20], decreasing the TEER values of the Caco-2 monoculture from 1300 ± 196 Ω·cm^2^ when they are seeded on flat membranes (Appendix A) to 140 ± 30 Ω·cm^2^ when cultured on 3D hydrogels (Figure 1E).

As HT29-MTX cells are known to modulate the geometry of the tight junctions [30,31], the presence of these goblet-like cells decreased the tightness of the barrier even more to 47 ± 21 Ω·cm^2^ (Figure 1E), perfectly mimicking the resistance values of the human small intestine reported to be in the range of 12–69 Ω·cm^2^ [32]. Unexpectedly, these lower TEER values for the 3D co-cultured sample did not correlate with an increase in the permeability of the model compound FD4. The diffusion of the dextran probe was significantly hindered when co-culturing Caco-2 and mucus-secreting HT29-MTX cells on the 3D scaffolds (Figure 1F). On the contrary, in the 2D Transwell inserts, the presence of goblet cells led to lower TEER measurements and, despite the mucus secretion to higher permeability values (Appendix A), as different authors previously reported [30,33,34]. In this 2D co-culture model the HT29-MTX cells are reported to grow on patches surrounded by the Caco-2 cells [35]. This would imply a mucus layer that is not uniformly distributed, and therefore a compromised mucus barrier. However, in vivo the intestinal mucus layer is known to act as a barrier to oral permeability of both hydrophilic and hydrophobic compounds [3,4].

**Figure 1 pharmaceutics-14-01552-f001:**
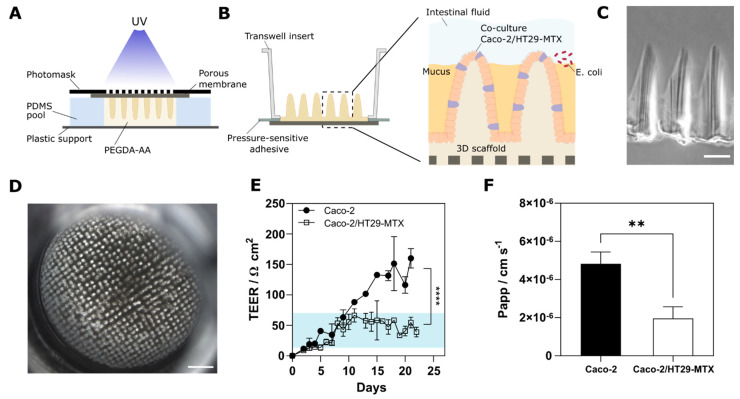
**Fabrication of the 3D villus-like microstructured models.** (**A**) Schematic drawing of the fabrication method of the villus-like hydrogel scaffold. (**B**) Schematic drawing of the villus-like hydrogel assembled into a conventional Transwell insert. The 3D biomimetic model of the small intestinal barrier contains the microstructured villus-like hydrogel scaffold; the epithelial cell barrier co-culturing the enterocyte-like Caco-2 cells and the goblet-like HT29-MTX cells; the secreted mucus layer where the host–pathogen interactions occur; and the intestinal fluid. (**C**) Brightfield microscope image of the cross-section of the villus-like hydrogel scaffold. Scale bar = 200 μm. (**D**) Top view image of the Caco-2/HT29-MTX co-culture on the villus-like hydrogel scaffold assembled on the Transwell insert after 21 days culture. Scale bar = 1 mm. (**E**) Transepithelial electrical resistance (TEER) of the 3D models along the days of culture (black dots: monoculture of Caco-2 cells; open squares: co-culture Caco-2/HT29-MTX). The blue shadow indicates the range of the reported TEER values of the human small intestine [32]. (**F**) Apparent permeability (*Papp*) values of the FD4 model compound across the 3D models. Mean ± SEM, *n* > 10, ** *p* < 0.01; **** *p* < 0.0001.

Our 3D scaffolds provided a physiologically relevant topography where the cells grew and polarized along the finger-like structures replicating the in vivo tissue morphology of intestinal villi (Figure 2A). The PAS staining revealed that these villus-like hydrogel scaffolds also enhanced mucus production, even when only Caco-2 cells were seeded (Figure 2B). Similar 3D intestinal tissue structures have been shown to induce mucin secretion of Caco-2 cells [16]. This thin mucus layer, that was more evident at the tips of the villi, may have some hindering effect on the diffusion of the dextran probe [16]. Conversely, in the co-culture 3D model, the secreted mucus formed a continuous thick layer (Figure 2B), reducing the permeability of FD4 significantly and thus better reproducing the intrinsic barrier properties of the native mucosal tissue. The villus-like 3D scaffolds also enhanced cell differentiation and polarity (Figure 2C) [20]. After 2 weeks of co-culture, epithelial cells showed a polarized morphology with a strong accumulation of F-actin in the apical membrane and a columnar shape. The immunostaining of the 3D co-culture model showed the characteristic polarization markers of the intestinal epithelial cells. β-catenin was localized in the basolateral membrane and lateral borders of the epithelial cells. The brush border protein villin was mainly localized in the apical cell membrane, and the tight junction protein ZO-1 accumulated sharply in the cell-cell junctions. The immunostaining of MUC5AC, the mucin predominantly secreted by the HT29-MTX cells, showed the distribution of these goblet-like cells along the villus microstructures (Appendix A).

These results demonstrate that our 3D villus-like model containing both enterocyte and goblet-like cells mimics the barrier properties of the intestinal mucosa, both in terms of the cellular barrier integrity and the mucus barrier to intestinal permeation, as well as the epithelial cell polarity.

### 3.2. The Mucus Layer Acts as a Protective Barrier against Bacterial Invasion

The mucus lining the intestinal epithelium is the first line of defense against pathogens preventing them from reaching the epithelium. Therefore, it is crucial to reproduce this mucosal barrier in our biomimetic 3D model to enable the study of the intestinal mucosa–pathogen interactions. Thus, we challenged our models with two *E. coli* strains: the non-pathogenic MG1655 and the pathogenic LF82. After 3 h of infection at MOI of 100, the bacterial adhesion was determined. The ability to invade the epithelial models was investigated using the gentamicin kill assay. As expected, the adherent-invasive *E. coli* strain LF82 had higher ability to adhere to intestinal epithelial cells than the control MG1655 strain (Figure 3A). In flat standard cultures, the presence of the mucus layer when co-culturing Caco-2 and HT29-MTX cells increased the bacterial adhesion significantly in both strains (Appendix A). However, in the 3D scaffolds, this adhesion rate was similar for both models.

While the PAS staining revealed that the co-culture model formed a thick mucus layer, the monoculture of Caco-2 cells in the 3D villus-like hydrogels also induced mucus secretion (Figure 2B). The glycoproteins of this thin mucus layer provided enough sites for bacteria to adhere [36], which might explain the results found. However, when we looked at the invasion rate, our 3D model evidenced that the mucus layer acted as an effective protective barrier against bacterial penetration. In this case, the thick and continuous mucus layer formed in the co-culture model significantly hindered the bacterial invasion (Figure 3B). It has been demonstrated that the invasive *E. coli* LF82 strain has the ability to cross the intestinal mucus by promoting mucin degradation with proteases [25] and modulating the expression of flagella [37]. In fact, we observed that the invasion rate of this pathogen was significantly higher than the commensal *E. coli* MG1655 (Figure 3B). However, our results showed that the invasion was still more efficient in the absence of mucus. Actually, compared to the 2D model where Caco-2 cells were completely devoid of mucus, the invasion rate of the 3D monoculture was 3.5-fold lower indicating that even the thin mucus layer secreted in the Caco-2 3D model had some protective effect against invasion (Appendix A). Our results are in agreement with the work of Kim and co-workers, where they showed that the infection of their 3D intestinal model of Caco-2 monocultures with Salmonella typhimurium was restricted to the crypt region of the scaffold devoid of mucus and when they suppressed the MUC17 expression, the invaded bacteria increased by 10-fold, penetrating both in the crypts and in the villi [16].

The presence of the mucus layer secreted in the 3D models was also confirmed by SEM (Figure 3C). Caco-2 cells on top of the 3D scaffold revealed a mucus secretion covering some parts of the epithelial surface and the bacteria partially entrapped (white arrows). The microvilli brush border of the differentiated enterocytes was clearly observed. In the co-culture model, the mucus droplets secreted by the goblet cells were scattered throughout the epithelial surface indicating a uniform distribution of these cells after 3 weeks of co-culture. In this case, the mucus blanket, which has a very similar structure to the native mucus [38], was fully colonized by bacteria.

Thus, our results evidenced the key role of the intestinal mucus barrier on bacterial infection. The mucus layer promotes the adhesion of bacteria while it acts as a protective barrier against invasion of the CD-related *E. coli* LF82 strain.

### 3.3. The Infection with Crohn’s Disease-Associated E. coli LF82 Shows Hallmarks of the Inflamed Intestinal Mucosa

Crohn’s disease (CD) is a chronic inflammatory disease that results from a complex interplay between genetic, environmental, and microbiota dysbiosis factors [39]. Patients are characterized by a compromised epithelial barrier function and an abnormal inflammatory response. CD-associated AIEC strains have the ability to induce intestinal inflammation by disrupting the intestinal barrier and replicating intracellularly [22,40,41]. Thus, we measured the barrier integrity and the bacterial survival within our 3D intestinal barrier models for the CD-associated LF82 and commensal MG1655 *E. coli* bacterial strains. For each strain, the number of intracellular bacteria after 24 h of gentamicin exposure was compared with that determined after 1 h of gentamicin treatment. Our results showed that the *E. coli* LF82 strain was able to survive inside the epithelial cells both in the 3D monoculture and the co-culture, whereas for the non-pathogenic MG1655 strain, the percentage of intracellular bacteria decreased significantly after 24 h (Figure 4A).

Even though the *E. coli* LF82 has been shown to survive and replicate inside non-polarized epithelial cells without damaging them, it leads to disruption of epithelial barriers and to defects in the junction proteins [22,41]. In our 3D models, the infection with LF82 led to a significant reduction in TEER compared to uninfected sham controls 24 h after the bacterial exposure (Figure 4B). Caco-2 monoculture showed a higher breach of the barrier integrity than the co-culture, as a result of the higher rate of invading bacteria. Another key pathogenic feature of AIEC bacteria is to trigger intestinal inflammation through the overproduction of proinflammatory cytokines such as IL-8 secreted by the epithelial cells and macrophages [24,42]. In CD patients, the AIEC LF82 strain has been shown to induce increased expression of IL-8, among other cytokines [43]. Thus, we measured the apical levels of this proinflammatory cytokine after infection. As expected, the LF82 induced a significant increase in the secretion of IL-8 compared to the control for both 3D models (Figure 4C). However, the inflammatory response in the Caco-2 monoculture was much more acute than the HT29-MTX-containing co-culture, again correlating with the results of invasion rate and barrier disruption.

Altogether, our results show that the mucus barrier acts as a protective barrier when studying the cell response against infection, highlighting again the importance of representing in vitro all the key players involved in the host–pathogen interactions.

### 3.4. The Luminal Microenvironment Modulates E. coli LF82 Invasion

The intestinal luminal fluid is known to be essential for the solubility and permeability of certain nutrients and drugs. The combination of secreted bile salts and lipids forms mixed micelles that solubilize lipophilic molecules and modulate drug absorption [44]. It is also known that the bile salts interact with the microbiota in a bidirectional manner. On the one hand, the gut microbiota contributes to the bile salts’ metabolism by generating secondary bile acids. On the other hand, bile salts shape the composition of the microbiota and modulate the pathogenicity of some strains [45]. Specifically, the CD-associated *E. coli* LF82 strain has been shown to increase pathogenicity in contact with bile salts [25,46]. Thus, in order to fully biomimic the luminal environment in which the host–pathogen interactions are taking place, we performed the infection experiments in simulated intestinal fluid (SIF). This SIF contained physiological concentrations of bile salts and lipids found in fasted conditions [47], in this case, 5 mM sodium taurocholate and 1.25 mM L-phosphatidylcholine. SIF did not affect the viability of the 3D intestinal co-culture model and neither did the SIF metabolized products (Appendix A). In order to promote the expression of virulence factors, LF82 was incubated overnight in SIF. This preincubation did not alter the kinetics of bacterial growth (Appendix A). The exposure of the *E. coli* pathogen LF82 to SIF did not lead to an increase in bacteria adhesion to the mucus-containing 3D model, but to an increase in pathogen invasion (Figure 5). Previous works demonstrated that the incubation of *E. coli* LF82 with high concentrations of bile salts induced the overexpression of some virulence factors such as the Vat-AIEC protease or the long polar fimbriae leading to an increase in the adhesion rate to mucus-producing cells [25,46]. In our model, the biorelevant SIF media favored overpassing the mucus layer, resulting in higher bacterial penetration.

Although further studies should be performed to elucidate the mechanisms of this increased invasiveness, our results demonstrate that in our 3D model, the AIEC strain responds to the luminal environment.

## 4. Discussion

The intestinal mucosa is a complex organ with multiple features that modulate its interactions with the microbiome. The cell population of the intestinal epithelium, the characteristic 3D topography, and the mucus barrier play a key role in how our body maintains homeostasis and responds to potential pathogens [48]. The intestinal epithelium is composed of a monolayer of multiple cell types distributed along with a crypt-villus 3D structure. The absorptive enterocytes and the goblet cells are the most abundant intestinal cells. Goblet cells are responsible for the secretion of mucus, which forms a dynamic protective layer limiting the influx of bacteria and bacteria antigens [7]. Other secretory cells such as Paneth cells located in the intestinal crypts also have a role in maintaining the equilibrium between the gut and the intestinal microorganisms by secreting anti-microbial compounds [49]. In the last decade, researchers have developed multiple experimental models to study and mimic in vitro the bacteria-host interactions in the gut [10,12]. In vitro systems based on intestinal organoids offer complex and elegant models containing all the intestinal cell types that self-organize into crypt and villus domains [13,50]. Although most of these studies have used organoids with a closed lumen as models [51], in the last years organoid-derived monolayers have been developed easing the access to the apical compartment and have been employed to study, for example, the invasive kinetics of intestinal pathogens [52,53,54]. However, these models fail to recapitulate the 3D crypt-villus axis characteristic of the small intestine and the mucus layer is not well represented [55]. This mucus layer is known to play a key role in the interactions between the microorganisms and the gut [56] and therefore, it constitutes an indispensable component in a representative in vitro model of the host–pathogen interactions of the intestinal mucosa. One of the most used models to study the intestinal mucus barrier is the co-culture of the enterocyte-like Caco-2 cells and the goblet-like HT29-MTX cells. This co-culture model has been used mainly in 2D using Transwell inserts to study the effect of the mucus layer on drug permeability [57]. The mixture of these two very well-established cell lines is known to reduce the tightness of the intestinal barrier, and their culture in air-liquid conditions enhances mucus secretion [58]. On the other hand, epithelial cells have been shown to respond to 3D topography. The pronounced curvature of the intestinal crypt-villus structures induces changes in the cell morphology [20,59] and in the secretion of mucus [16]. Thus, in this work we developed a hydrogel-based 3D in vitro model of the intestinal mucosa with the characteristic villus-like topography, the intestinal epithelium, and the mucus layer to study the role of this mucosal barrier in the adhesion and invasion of the AIEC pathogen LF82. In this 3D model, the co-culture of Caco-2/HT29-MTX cells was arranged in a polarized monolayer along the villus microstructures showing in vivo-like TEER values.

The presence of the goblet cells together with the 3D topography that enhanced the mucus secretion [16,60] led to the formation of a thick mucus layer with effective barrier properties that hindered the diffusion of the FD4 model compound. This mucus layer also induced a protective effect against bacterial invasion, both for the commensal MG1655 and for the pathogenic LF82. The invasive ability and the invasion mechanisms of the CD-associated LF82 strain have been thoroughly studied, mostly using mucus-free models of non-polarized intestinal epithelial cells [21,22,40,41]. LF82 adheres to the CEACAM-6 receptor of the intestinal epithelial cells and induces epithelial barrier disruption [41,42]. However, for the pathogen to adhere to the epithelial surface, first it has to overcome the intestinal mucus barrier. It has been demonstrated that the LF82 strain has different strategies to penetrate the mucus layer: the secretion of the Vat-AIEC protease that degrades the mucus, and the expression of flagella that promotes motility and bacterial adhesion [25,37]. Despite these competitive advantages, it is not surprising that the thick mucus layer observed in our 3D co-culture model offered an increased protection against this pathogenic invasion compared with the Caco-2 monoculture model. Unlike patients with ulcerative colitis in which the mucus layer is greatly disrupted in the inflammation area, patients with CD show a continuous mucus layer with thicknesses comparable to the healthy controls. However, this mucus shows an altered structure that leads to a decrease in viscoelasticity and reduced barrier function [61]. This aberrant mucus structure and the mucinolytic activity reported for the AIEC strain favored gut colonization and induce inflammation in CD [24]. In addition, the expression of the Vat-AIEC protease is known to be influenced by the intestinal luminal microenvironment [25], so the exposure of the *E. coli* LF82 strain to a biorelevant media (SIF) enhanced further this invasion ability. After infection, the LF82 was able to survive and replicate inside the epithelial cells of our models, causing a barrier disruption. This interaction of the AIEC strain with the intestinal epithelial cells induced an overproduction of the proinflammatory cytokine IL-8. This inflammatory response was also affected by the presence of mucus. Although the IL-8 release by *E. coli* LF82 infection was 5-fold higher than the control in the 3D co-culture model, this response was much less acute than in the 3D Caco-2 model. The thick mucus layer not only protected the epithelium from invasion, but also from the molecular interactions that trigger the proinflammatory cytokine release and therefore damage the cells.

The barrier effect of the intestinal mucus layer to bacteria penetration has also been investigated for other bacterial strains. Sharma and co-workers showed that non-motile probiotic bacteria were significantly entrapped in a mucus gel layer whereas the penetration of the flagellated *Salmonella* pathogen after 12 h infection was even enhanced when the mucus layer was present in the model [62]. These results differ from the ones obtained by Kim et al. [16]. They found that the bacterial penetration of the flagellated *Salmonella* strain *S. typhimurium* was significantly reduced in their villus-like 3D model, which highly expressed MUC17. Other studies on the motility of this strain in mucus revealed that infection predominantly occurred at sites featuring some gaps in the mucus layer [63].

On the other hand, the 3D topography of the small intestine has also been shown to have an effect on bacterial infection. Alzheimer and co-workers demonstrated that the adhesion and transmigration of *C. jejuni* was delayed and decreased when performed in their 3D Caco-2 model compared to the standard 2D Caco-2 model [64]. Altogether, these results reinforced our premise that to recapitulate better the host–bacterial interactions in the gut, it is essential to include the multiple features that modulate those interactions: the 3D topography, the intestinal epithelium, and the mucus layer, in a biorelevant microenvironment. Additional complexity could be introduced in the model in the form of an immune cell compartment or a dynamic environment to advance our understanding of the interactions of the intestinal microorganisms with the mucosal barrier.

## 5. Conclusions

In this work we developed an in vitro model that faithfully mimics the human small intestinal mucosa, including the characteristic 3D villus topography, the epithelial barrier, the secreted mucus layer, and the luminal microenvironment. The study of the interactions with the Crohn’s disease-associated strain LF82 revealed that the mucus layer acts as an important protective layer against bacterial invasion, also mitigating the cell response upon infection. These host–pathogen interactions can also be influenced by the exposure to the luminal intestinal fluid.

## Figures and Tables

**Figure 2 pharmaceutics-14-01552-f002:**
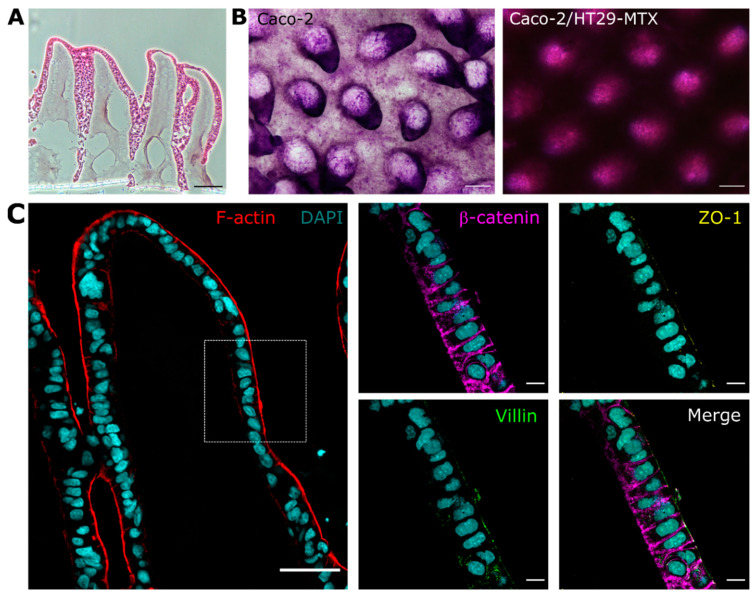
**Characterization of the 3D villus-like co-culture model.** (**A**) HE staining of the 3D villus-like model with the co-culture of Caco-2/HT29-MTX cells after 2 weeks of culture. Scale bar = 100 μm. (**B**) Periodic acid-Schiff (PAS) staining of the monoculture of Caco-2 (**left**) or the Caco-2/HT29-MTX co-culture (**right**) after 2 weeks of culture. Scale bars = 200 μm. (**C**) Confocal images of the cross-section of Caco-2/HT29-MTX co-culture grown on top of the villus-like microstructures for 2 weeks. F-actin is shown in red and nuclei in cyan. Scale bar = 50 μm. The doted box region is shown in high magnification (right panels). The polarization marker β-catenin is shown in magenta, ZO-1 in yellow, and villin in green. Scale bars = 10 μm.

**Figure 3 pharmaceutics-14-01552-f003:**
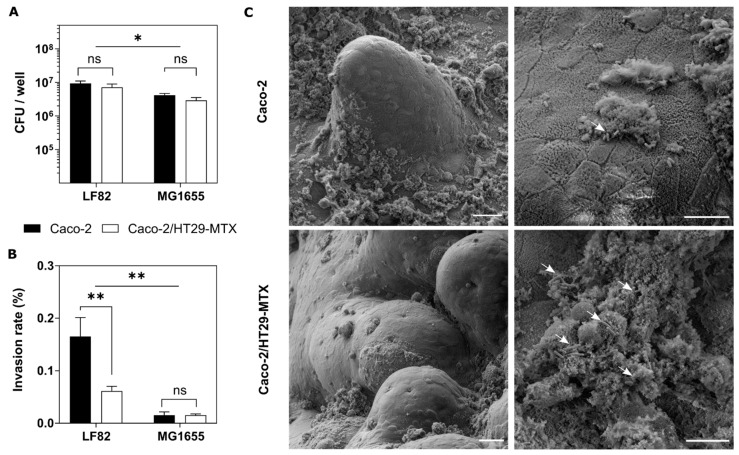
**Adhesion and invasion assays.** Quantification of adherent (**A**) or invaded (**B**) bacteria (*E. coli* LF82 or MG1655 strains) to the 3D Caco-2 monoculture (black bars) or the 3D Caco-2/HT29-MTX co-culture (white bars) after 3 h of infection. The bacterial adhesion is expressed in CFU/well. The bacterial invasion is determined after 3 h of infection and subsequent incubation with gentamicin solution for 90 min. Invasion is expressed as the percentage of invaded *E. coli* relative to the adhered bacteria. Mean ± SEM of at least two independent experiments with 3 replicates, ns (*p* > 0.05), * (*p* < 0.05), ** (*p* < 0.01). (**C**) Scanning electron microscopy (SEM) images of the 3D Caco-2 monoculture (upper panels) or the 3D Caco-2/HT29-MTX co-culture (lower panels) infected with *E. coli* LF82 for 3 h. White arrows indicate bacteria entrapped in the mucus. Scale bars = 25 μm (**left** panels) and 10 μm (**right** panels).

**Figure 4 pharmaceutics-14-01552-f004:**
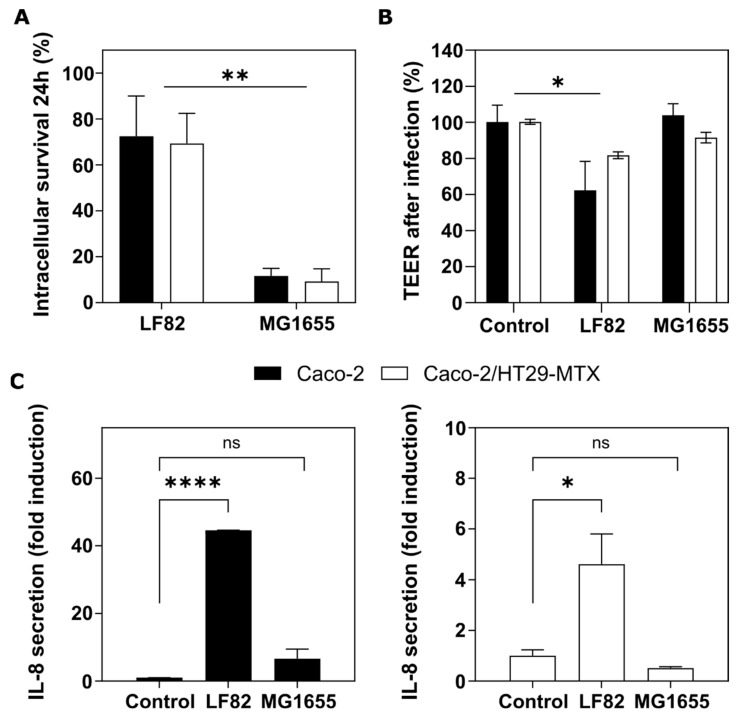
**Cellular response to bacterial infection.** (**A**) Survival of the *E. coli* LF82 or MG1655 strains within the 3D Caco-2 monoculture (black bars) or the 3D Caco-2/HT29-MTX co-culture (white bars) for 24 h. The intracellular survival is calculated as the percentage of intracellular bacteria relative to the number after 1 h of gentamicin treatment. (**B**) Transepithelial electrical resistance (TEER) after 24 h of gentamicin treatment relative to the values before infection. Mean ± SEM of at least two independent experiments with 3 replicates. (**C**) IL-8 release in the apical compartment of the 3D Caco-2 monoculture (black bars, left panel) or the 3D Caco-2/HT29-MTX co-culture (white bars, right panel) after 24 h of gentamicin treatment. Mean ± SEM of two replicates, ns (*p* > 0.05), * (*p* < 0.05), ** (*p* < 0.01), **** (*p* < 0.0001).

**Figure 5 pharmaceutics-14-01552-f005:**
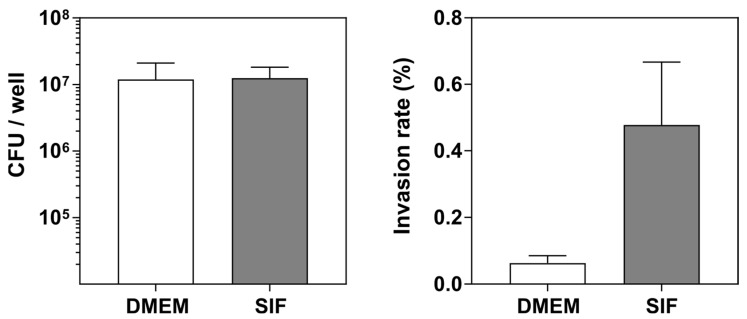
**Simulated intestinal fluid.** Quantification of the adhered (**left** panel) and invaded (**right** panel) bacteria to the 3D co-culture model infected with the pathogenic *E. coli* LF82 for 3 h after on overnight incubation with DMEM (white bars) or simulated intestinal fluid (SIF, grey bars). Mean ± SEM of two independent experiments with 3 replicates.

## Data Availability

The raw data supporting the conclusions of this article will be made available by the authors upon request.

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
