# Peer review of "Mimicking the Intestinal Host–Pathogen Interactions in a 3D In Vitro Model: The Role of the Mucus Layer"

_pharmaceutics, 2022, doi:10.3390/pharmaceutics14081552_

Round 1
Reviewer 1 Report
Dear Authors,
This is a very interesting paper as it presents a useful and easy-to-create tool for studying many aspects of the IBD pathogenesis, as well as, the host-microbiota interactions. Some minor comments from my part:
1. Since the authors co-cultured two different epithelial cell lines, I believe it would be interesting to stain the 3D mixed structures in order to pinpoint the distribution of these cells.
2. Both of these two epithelial cell lines have derived from intestinal cancerous tumors and thus, their characteristics may differ from the normal epithelium. I believe that it would be useful for the authors to compare their model with epithelial cells isolated from healthy intestinal mucosa.
3. The authors mention that they cultured their 3D model for approximately 3 weeks. How did they expand their model? Did they reseed the cells in new scaffolds when their model was overpopulated with the epithelial cells? If yes, how did they evaluate that their 3D model kept its characteristics each time they reseeded the cells?
Author Response
- Since the authors co-cultured two different epithelial cell lines, I believe it would be interesting to stain the 3D mixed structures in order to pinpoint the distribution of these cells.
We thank the reviewer for his/her comments. We have now included a supplementary figure with an immunostaining for MUC5AC, that is the mucin that the HT29-MTX cells secrete. This figure contains different sections of a villus-like microstructure along the z-axis that show the distribution of the goblet cells in the 3D co-culture model.
- Both of these two epithelial cell lines have derived from intestinal cancerous tumors and thus, their characteristics may differ from the normal epithelium. I believe that it would be useful for the authors to compare their model with epithelial cells isolated from healthy intestinal mucosa.
We agree that having primary cells isolated from healthy intestinal mucosa would be very interesting, and we would consider the suggestion for the next steps in our research. But we started the model with very well stablished cell lines, to minimize the complexity of the model and to be able to evaluate the effect of the intestinal mucus and luminal microenvironment on the infection of a pathogenic E. coli strain.
- The authors mention that they cultured their 3D model for approximately 3 weeks. How did they expand their model? Did they reseed the cells in new scaffolds when their model was overpopulated with the epithelial cells? If yes, how did they evaluate that their 3D model kept its characteristics each time they reseeded the cells?
We apologize for the misleading statement. The culture of the 3D model goes as follows: Before seeding the cells on the scaffolds, the individual Caco-2 and HT29-MTX cell lines are expanded and maintained separately in standard culture flasks. For each experiment, cells are mixed in the 10:1 Caco-2/HT29-MTX ratio and seeded on top of the scaffolds. These cells need a few days to colonize and cover the full 3D villus-like scaffold, and then 2 to 3 weeks to differentiate and polarize into enterocyte- or goblet-like cells, that we maintained in air-liquid interface conditions. During this 3-week culture, we just changed the media every other day. After that, the infection experiments were performed. We have now modified the methods’ section to explain this procedure better.
Reviewer 2 Report
The manuscript “Mimicking the intestinal host-pathogen interactions in a 3D in vitro model: the role of the mucus layer” addresses a relevant topic: addresses an important issue: the implementation of in vitro cell models that mimic the in vivo situation as closely as possible. By using a well-characterized co-culture cell model for the intestinal tract including mucus and fine-tuning it by introducing a 3D hydrogel structure that mimics the brush border of the intestinal tract, this study introduces a novel in vitro intestinal cell model. I believe that this model will be highly appreciated by scientists in the field and will generate interest among readers of Pharmaceutics.
Author Response
We thank the reviewer for her/his positive comments.
Reviewer 3 Report
It's a good idea to develop in vitro model that mimics the small intestinal mucosa and investigate interactions between intestinal pathogens and intestinal mucosa. This study would be useful information to elucidate other interactions occur in the small intestine. Moreover, this manuscript has been written to be well understood. However, I bring some minor comments to the authors' attention.
1. In the small intestine, there is a Payer's patch that is closely related to the absorption of foreign substances and the occurrence of diseases. Based on the model presented in this study, what do you think about modeling using cells (macrophages, etc.) and co-culture?
2. What do you think about adding an enzyme secreted from the small intestine (trypsin) to SIF?
Author Response
- In the small intestine, there is a Payer's patch that is closely related to the absorption of foreign substances and the occurrence of diseases. Based on the model presented in this study, what do you think about modeling using cells (macrophages, etc.) and co-culture?
The authors thank the reviewer for his/her comment. We agree that there are other cells in the small intestine, such as M cells or Paneth cells, that have an important role in the immune surveillance of the intestinal tissue, as well as the immune cells located in the stromal compartment. However, in this work, we focused on the two most abundant cells of the intestinal epithelium (enterocytes and goblet cells) to study the effect of the intestinal mucus on bacteria adhesion and invasion, using a simple but physiologically relevant model. This model can be adapted in future studies to include other types of cells relevant to the host-pathogen interactions.
- What do you think about adding an enzyme secreted from the small intestine (trypsin) to SIF?
The authors thank the reviewer for his/her suggestion. There are different compositions of the simulated intestinal fluid depending on the conditions (preprandial or postprandial) that are mimicking. Fasted state (FaSIF) and fed state (FeSIF) fluids differ in the concentration of bile salts and lipids (higher in FeSIF), and the presence of the pancreatin (trypsin) enzymes in the FeSIF. In this first approach we wanted to investigate the effect of the luminal microenvironment on E. coli LF82 infection in basal (fasted) conditions. We have now specified it in the manuscript. In future studies we can include the FeSIF to have a complete picture of the effect of the luminal fluid on the interaction of pathogens in the small intestine.
Reviewer 4 Report
Comments to the Author:
1. Could you explain more the motivation for making 3D Caco-2/HT29-403 MTX co-culture? What’s the advantage?
2. What’s the power-value of the Figure 4?
3. Do you have more explanations for Figure 2? What’s the significance between the difference of fluorescent cell images in Figure 2?
4. There are several issues with the grammar and English. Please revise the manuscript thoroughly.
6. A large body of latest literature on in vitro culture using gut organoids is missing and must be discussed and referenced in their work. This is important so that readers are aware of the current literature. They are:
1) Gut Organoid as a New Platform to Study Alginate and Chitosan Mediated PLGA Nanoparticles for Drug Delivery. Marine Drugs 2021, 19(5): 282-298.
2) Manipulate Intestinal Organoids with Niobium Carbide Nanosheets. Journal of Biomedical Materials Research Part A 2021, 109(4): 479-487.
3) Transport of Artificial Virus-like Nanocarriers (AVN) through Intestinal Monolayer via Microfold Cells. Nanoscale 2020, 12(30): 16339-16347.
4) Ex vivo Study of Telluride Nanowires in Minigut. Journal of Biomedical Nanotechnology 2018, 14(5): 978-986.
5) Effects of Six Common Dietary Nutrients on Murine Intestinal Organoid Growth. PLoS ONE 2018, 13(2): e0191517.
6) Intestinal Organoids Containing PLGA Nanoparticles for the Treatment of Inflammatory Bowel Diseases. Journal of Biomedical Materials Research Part A 2018, 106(4): 876-886.
Author Response
1. Could you explain more the motivation for making 3D Caco-2/HT29-403 MTX co-culture? What’s the advantage?
Caco-2 is the most frequently used cell line for modeling healthy intestinal interactions, as it recapitulates some of the features of the intestinal enterocytes and their interactions with intestinal microbes. Also, this cell line is considered the gold standard model for intestinal testing, being approved by the FDA. Since one of the objectives of this work was to evaluate the role of the intestinal mucus, the co-culture with HT29-MTX cells was included. This Caco-2/HT29-MTX co-culture is one of the most used models to study the intestinal mucus barrier mainly in 2D using Transwells. Our work demonstrated that the use of this Caco-2/HT29-MTX co-culture on top of 3D villus-like scaffold (i) better recapitulated the intestinal barrier properties (TEER and Papp); and (ii) enhanced the mucus secretion, providing a robust model of the intestinal mucosa to study host-pathogen interactions in a physiologically relevant manner. The advantages of using these cell lines are their robustness, ease of use, and reproducibility. We have now added the following sentences to the introduction: “The Caco-2 is the most used cell line for modeling the small intestine and the gold standard model for permeability drug screening; whereas its co-culture with HT29-MTX cells allows for the secretion of mucins predominantly expressed in the intestinal tract.”
2. What’s the power-value of the Figure 4?
We thank the reviewer for his/her suggestion. We have now added the statistic analysis to Figure 4.
3. Do you have more explanations for Figure 2? What’s the significance between the difference of fluorescent cell images in Figure 2?
The fluorescent images in Figure 2 are immunostainings of different markers characteristic of the polarized intestinal epithelium. The formation of a continuous apical structure positive for actin and villin is typically considered as a criterium for epithelial polarization (Baas et al. Cell (2004); vol 116, 457-466). Moreover, we also stained for b-catenin, that accumulates in the basolateral membrane and lateral borders of polarized epithelial cells, and ZO-1, which is a typical marker of tight junction proteins in the intestinal epithelium. The manuscript has been modified to include these explanations.
4. There are several issues with the grammar and English. Please revise the manuscript thoroughly.
The text has been thoroughly revised.
5. A large body of latest literature on in vitro culture using gut organoids is missing and must be discussed and referenced in their work. This is important so that readers are aware of the current literature.
They are:
1) Gut Organoid as a New Platform to Study Alginate and Chitosan Mediated PLGA Nanoparticles for Drug Delivery. Marine Drugs 2021, 19(5): 282-298.
2) Manipulate Intestinal Organoids with Niobium Carbide Nanosheets. Journal of Biomedical Materials Research Part A 2021, 109(4): 479-487.
3) Transport of Artificial Virus-like Nanocarriers (AVN) through Intestinal Monolayer via Microfold Cells. Nanoscale 2020, 12(30): 16339-16347.
4) Ex vivo Study of Telluride Nanowires in Minigut. Journal of Biomedical Nanotechnology 2018, 14(5): 978-986.
5) Effects of Six Common Dietary Nutrients on Murine Intestinal Organoid Growth. PLoS ONE 2018, 13(2): e0191517.
6) Intestinal Organoids Containing PLGA Nanoparticles for the Treatment of Inflammatory Bowel Diseases. Journal of Biomedical Materials Research Part A 2018, 106(4): 876-886.
We thank the reviewer for his/her comment. We agree that new models of the intestinal mucosa have been developed in the recent years using intestinal organoids, not only to study host-pathogen interactions but also for drug delivery and disease modeling. The references suggested by the reviewer are surely relevant for the development of new treatment strategies using drug delivery systems in diseases such as IBD. However, since our work mainly focused on the microbiota-cell interactions of the intestinal mucosa and the role of the mucus layer in those interactions, we narrowed the manuscript’s discussion and state-of-the-art to the literature relevant in this specific field. We agree with the reviewer that there were some important works on the use of intestinal organoids to study host-pathogen interactions missing, therefore we have now included the following references:
https://doi.org/10.1016/j.chom.2021.04.002
https://doi.org/10.1053/j.gastro.2014.09.042
https://doi.org/10.1128/Spectrum.00003-21
https://doi.org/10.3389/fcimb.2020.00272
Round 2
Reviewer 4 Report
All the previous concerns have been properly addressed. This manuscript is acceptable in the current version.